# What Does Bone Corticalization around Dental Implants Mean in Light of Ten Years of Follow-Up?

**DOI:** 10.3390/jcm11123545

**Published:** 2022-06-20

**Authors:** Marcin Kozakiewicz, Małgorzata Skorupska, Tomasz Wach

**Affiliations:** Department of Maxillofacial Surgery, Medical University of Lodz, 113 Zeromskiego Str., 90-549 Lodz, Poland; malgorzata.glonek@umed.lodz.pl (M.S.); tomasz.wach@umed.lodz.pl (T.W.)

**Keywords:** dental implants, long-term results, long-term success, marginal bone loss, functional loading, intraoral radiographs, radiomics, texture analysis, corticalization, bone remodeling

## Abstract

The phenomenon of peri-implant bone corticalization after functional loading does not yet have a definite clinical significance and impact on prognosis. An attempt was made to assess the clinical significance of this phenomenon. This prospective study included 554 patients. Standardized intraoral radiographs documenting the jawbone environment of 1556 implants were collected. The follow-up period was 10 years of functional loading. Marginal alveolar bone loss (MBL) and radiographic bone structure (bone index, BI) were evaluated in relation to intraosseous implant design features and prosthetic work performed. After five years, bone structure abnormalities expressed by a reduction of BI to 0.47 ± 0.21 and MBL = 0.88 ± 1.27 mm were observed. Both values had an inverse relationship with each other (*p* < 0.0001). Reference cancellous bone showed BI = 0.85 ± 0.18. The same relationship was observed after ten years of functional loading: BI = 0.48 ± 0.21, MBL = 1.49 ± 1.94 mm, and again an inverse relationship (*p* < 0.0001). Increasing corticalization (lower BI) is strongly associated with increasing marginal bone loss and increasing corticalization precedes future marginal bone loss. Marginal bone loss will increase as corticalization progresses.

## 1. Introduction

It has been noted that after insertion of dental implants into living bone, the condition of the bone surrounding the implant changes with time [1]. Implant loading phenomena can induce bone remodeling. This is a fundamental behavior of normal bone [2,3,4]. This remodeling process consists of bone formation and resorption-dependent responses and activation signals toward the loading stimulus [5,6]. The remodeling of cortical bone by basal multicellular units occurs by a local osteoclastic resorption of osteons. Subsequently, the next generation of osteons is formed by osteoblast activity [7]. Functional mechanical loading has a direct effect on remodeling, which arranges the osteons along stress lines in the bone [8,9]. However, the question remains how the remodeling of peri-implant cancellous bone occurs.

This change refers to an increase in the optical density of the cancellous bone surrounding the dental implant by which it becomes similar to the cortical bone (Figure 1). Some authors refer to this process as peri-implant bone remodeling or mineralization [10]. However, it seems that these two terms do not describe the complexity of this tissue alteration. Hence, it is better to call it osseocorticalization or simply corticalization.

Assuming that the alveolar bone is the basis for the dental implants function according to their purpose in the act of biting and mastication [11,12], other factors such as soft tissue condition, depth of gingival pockets, gingival biotype, width of the keratinized gingival zone, color and translucency of soft tissues [13,14,15] are, however, secondary. In recent years, increasing attention has been paid to the phenomenon of corticalization [16,17,18]. It had been hypothesized that after bone remodeling of bone above the implant neck, almost no bone loss could be expected [19]. It is interesting whether this phenomenon affects the dimensions of the bone supporting a dental implant. The question arises whether this process is related to the vertical change of the bone level and the relationship to the prognosis of the dental implantation.

At the current stage of development of dental implantology, when uncertainty about the possibility of osseointegration of the implant has long been overcome, novel premises of success need to be looked for. Undoubtedly, one of the most important areas of examination is that bone surrounding the implant, and it is precisely these studies that are undertaken in this publication. An originality of our approach to this issue is the use of radiomics elements, i.e., a digital texture analysis [20].

The aim of this study was to evaluate long-term peri-implant jawbone corticalization and to try to indicate its clinical significance.

## 2. Materials and Methods

This was a prospective study based on a standardized collection of intraoral radiographs of 1556 dental implants. The inclusion criteria were: at least 18 years of age, healthy soft tissue (bleeding on probing <20%, plaque index <25%, community periodontal index for treatment needs <2), good oral hygiene, regular follow-ups and following doctor’s orders. The exclusion criteria were: uncontrolled internal comorbidity, a history of oral radiation therapy, past or current use of cytostatic drugs and low quality or lack of follow-up radiographs. Finally, clinical and radiological data of 554 persons were included into this study.

The dental implants were inserted by one dentist (M.K.) according to the protocols recommended by the manufacturers. Twenty-two types of dental implants were used in this study: AB Dental Devices I5 (www.ab-dent.com, Ashdod, Israel; accessed on 5 March 2022) 66 pieces, ADIN Dental Implants Touareg (www.adin-implants.com, Afula, Israel; accessed on 5 March 2022) 67 pieces, Alpha Bio ARRP (www.alpha-bio.net, Petah-Tikva, Israel; accessed on 5 March 2022) 9 pieces, Alpha Bio ATI (www.alpha-bio.net, Petah-Tikva, Israel; accessed on 5 March 2022) 73 pieces, Alpha Bio DFI (www.alpha-bio.net, Petah-Tikva, Israel; accessed on 5 March 2022) 27 pieces, Alpha Bio OCI (www.alpha-bio.net, Petah-Tikva, Israel; accessed on 5 March 2022) 16 pieces, Alpha Bio SFB (www.alpha-bio.net, Petah-Tikva, Israel; accessed on 5 March 2022) 48 pieces, Alpha Bio SPI (www.alpha-bio.net, Petah-Tikva, Israel; accessed on 5 March 2022) 254 pieces, Argon K3pro Rapid (www.argon-dental.de, Bingen am Rhein, Germany; accessed on 5 March 2022) 52 pieces, Bego Semados RI (www.bego-implantology.com, Bremen, Germany; accessed on 5 March 2022) 2 pieces, Dentium Super Line (www.dentium.com, Gyeonggi-do, South Korea; accessed on 5 March 2022) 10 pieces, Friadent Ankylos C/X (www.dentsplysirona.com, Warszawa, Poland; accessed on 5 March 2022) 9 pieces, Implant Direct InterActive (www.implantdirect.com, Thousand Oaks, CA, United States of America; accessed on 5 March 2022) 18 pieces, Implant Direct Legacy 3 (www.implantdirect.com, Thousand Oaks, CA, United States of America; accessed on 5 March 2022) 28 pieces, MIS BioCom M4 (www.mis-implants.com, Bar-Lev Industrial Park, Israel; accessed on 5 March 2022) 4 pieces, MIS C1 (www.mis-implants.com, Bar-Lev Industrial Park, Israel; accessed on 5 March 2022) 230 pieces, MIS Seven (www.mis-implants.com, Bar-Lev Industrial Park, Israel; accessed on 5 March 2022) 571 pieces, MIS UNO One Piece (www.mis-implants.com, Bar-Lev Industrial Park, Israel; accessed on 5 March 2022) 22 pieces, Osstem Implant Company GS III (www.en.osstem.com, Seoul, South Korea; accessed on 5 March 2022) 12 pieces, SGS Dental P7N (www.sgs-dental.com, Schaan, Liechtenstein; accessed on 5 March 2022) 6 pieces, TBR Implanté (www.tbr.dental, Toulouse, France; accessed on 5 March 2022) 6 pieces and Wolf Dental Conical Screw-Type (www.wolf-dental.com, Osnabrück, Germany; accessed on 5 March 2022) 26 pieces. No machined surface implant were used in this study—only rough surfaces. The tested implants are shown in Table 1. Healing was always unloaded and occluded in two-stage implants. Implant exposure was performed after 3 months, and functional loading was performed at that time. Fixed protheses were cemented.

Two-dimensional intraoral radiographs were taken with a Digora Optime system (Soredex, Helsinki, Finland). The radiographs were taken in a standardized way [21,22] with parameters: 7 mA, 70 mV an 0.1 s (Focus apparatus-Instrumentarium Dental, Tuusula, Finland). Positioners were used to take images repeatably with a 90° angle of the X beam to the surface of the phosphor plate. The texture of X-ray images was analyzed in MaZda 4.6 software invented by University of Technology in Lodz [23] to check how the features were changing through the 10 years of observation. A peri-implant’s region of interest (ROI) was established at both implant sites at the level of the implant neck/head. It had a 5 mm height. A reference ROI was located in the trabecular alveolar bone distant from the implant body (Figure 2). Data were collected in three time intervals: 1. initially when prosthetic work begins, 2. five years after functional loading of the implant, and 3. ten years after functional loading. The region of interests (ROIs) were normalized (*μ* ± 3*σ*) to share the same average (*μ*) and standard deviation (*σ*) as those of the optical density within the ROI. Selected image texture features (difference entropy from the co-occurrence matrix, and long-run emphasis moment from the run-length matrix) in the ROIs were calculated for the reference bone and peri-implant marginal bone:
(1)$$\mathrm{DifEntr}=-\sum_{i=1}^{Ng}p_{x-y}\left(i\right)log\left(p_{x-y}\left(i\right)\right)$$
where Σ is a sum, *Ng* is the number of levels of optical density in the radiograph, *i* and *j* are the optical density of pixels 5 pixels distant from another, *p* is probability, and *log* is the common logarithm [24]. The differential entropy calculated in this way is a measure of the overall scatter of bone structure elements in an X-ray image. Its high values are typical for cancellous bone [25,26,27].
(2)$$\mathrm{LngREmph}=\frac{\sum_{i=1}^{Ng}\sum_{k=1}^{Nr}k^{2}p\left(i,k\right)}{\sum_{i=1}^{Ng}\sum_{k=1}^{Nr}p\left(i,k\right)}$$
where Σ is a sum; *Nr* is the number of series of pixels with density level *i* and length *k*; *Ng*—number of levels for image optical density; *Nr*—number of pixel in series; and *p* is a probability [28,29]. This texture feature describes thick, uniformly dense, radio-opaque bone structures in intraoral radiograph images [25,26]. These two equations were subsequently used for the index construction [26,30]. The bone index (BI), which represent the ratio of the measure of the diversity of the structure observed in the radiograph to the measure of the presence of uniform longitudinal structures, was calculated by:(3)$$\mathrm{Bone}\;\mathrm{Index}=\frac{\mathrm{DifEntr}}{\mathrm{LngREmph}}=\frac{(-\sum_{i=1}^{Ng}p_{x-y}\left(i\right)log\left(p_{x-y}\left(i\right)\right))\sum_{i=1}^{Ng}\sum_{k=1}^{Nr}p\left(i,k\right)}{\sum_{i=1}^{Ng}\sum_{k=1}^{Nr}k^{2}p\left(i,k\right)}$$

Next, there was a measured shift of the marginal bone level at the implant neck assuming a reference level based on the observed level on radiographs taken at the time the prosthetic work was started. The study was repeated on radiographs 5 and 10 years after the functional loading of the implants (marginal bone loss, MBL).

A Kruskal–Wallis test was used for comparing the medians between variables with different qualitative features and a simple regression for confirming the existence of a relationship between two quantitative variables (Statgraphics-StatPoint Technologies, Inc., The Plains, VA, USA).

## 3. Results

Progressive corticalization in peri-implant bone was observed. It was dependent on time, implant design and type of prosthetic restoration. Marginal alveolar bone loss was also noted to occur during each study period. The results are shown in Figure 2, Figure 3 and Figure 4 and Table 2, Table 3 and Table 4.

A progressive change in the marginal bone structure was observed (Figure 3). This consisted in an increase in the number of longitudinal structures (increasing LngREmph) and a loss of the natural texture pattern characteristic of cancellous bone (decreasing chaotic patterns expressed by a decreasing differential entropy, DifEntrp). To determine the significance of the corticalization, it was related to the marginal bone loss. There was a statistical relationship (Figure 4 and Table 2) of structural changes with the amount of marginal bone loss. Moreover, even structure change preceded future bone loss (Figure 4b).

When evaluating the effect of the titanium alloy used for the implants, it was noted that the grade 4 alloy was associated with less corticalization of the surrounding bone after five and ten years of functional loading. At the same time, grade 4 was associated with less peri-implant bone loss (Table 3).

A subcrestal placement of implants gave the smallest radiological corticographic effect in the surroundings five years after the prosthesis has been installed on them. After a further five years of functional loading, bone level implants joined the subcrestal implants in a good way. In both distant observation periods, the greatest corticalization was caused by these tissue level implants (i.e., in this study, one-piece implants). The lowest marginal bone loss occurred when treated with subcrestal embedded implants.

When comparing the internal connection of the abutment with the abutment milled together with the implant (one-piece implant), it can be seen that less bone corticalization occurred in the internal connection group. However, this did not result in significantly less bone loss.

In terms of connection shape, the smallest amount of corticalization was found in the bone surrounding abutments constructed with a Morse cone and an internal hexagon. At the same time, little bone loss was detected when conical abutments were used.

The presence of microthreads within the implant head was not important to differentiate alveolar bone status. Both corticalization progressed gradually in both types of performance of this part of the implant, as well as in marginal bone loss over a five- and ten-year horizon.

It was noted that buttress threads, square threads and threadless implants were affected by the greatest peri-implant marginal bone loss. The buttress thread and threadless implants were associated with peri-implant bone hypercorticalization. Threadless (cylindrical) implants in particular caused a significant and rapid corticalization of the surroundings due to severe bone loss.

A significant dependence of the marginal bone remodeling on the applied prosthetic solutions was noticed. This resulted in varying degrees of marginal bone loss (Table 4).

## 4. Discussion

Obtaining primary stabilization is the first step to successful implant treatment [12,13]. It undoubtedly depends on the condition of the patient’s skeleton [31], but the surgical procedure of inserting the dental implant itself changes the bone, e.g., condenses it [32]. Corticalization can be found in very early period after implant placement [17,33]. Subsequently, the peri-implant bone structure changes depending on many factors [34,35,36], but inexorably increases its density around functionally loaded dental implants (Figure 3).

The long-term observations presented here were consistent with the data described by Gandolfi’s team in 2018: the amount of corticalization increased with the duration of dental implant use [37]. In addition, they examined bone in the implant neck region (as in this study) and comparatively in the apical region of the implant. Thus, it is known that corticalization is greater in the cortical portion of the implant. Presumably, this reduces the amount of space available for bone blood supply in the peri-implant bone at the level of neck implant portion, which may predispose it to the development of atrophy at five and ten years after functional loading (Figure 1 and Figure 2).The corticalization mechanism probably initially involves the process of consolidation of the implant surface with the alveolar crest bone osseointegration [13], which, after functional loading of the implants, causes an increase in the bone–implant contact area [38] by 12%. Considering that an implant used for a decade may have a fairly low level of bone contact, e.g., 30%, this 12% increase is clinically very important.

Furthermore, implants with rough surfaces are known to interact more with the host bone by increasing the bone-to-implant contact area. Tumedei et al. [38] reported that the increase was of 10%. Implants with machined surfaces were not studied at all in the series presented here.

When corticalization occurred (low bone index values), there was a strong association of corticalization (low BI = 0.41 ± 0.19, Figure 4a) present in the fifth year of implant use with marginal bone loss at that time (*p* < 0.0001). Even worse, it was also a predictor of marginal bone loss over the next 5 years (Figure 4b). Excessive corticalization of the bone directly in contact with the implant is unfavorable because it is associated with marginal bone loss after five and ten years of functional loading. High corticalization (low BI = 44 ± 0.18, Table 2) in the fifth year of implant loading was associated with bone loss in the tenth year of dental implant use (*p* < 0.0001). This is a prediction based on measuring the amount of bone corticalization at the implant neck. Obviously, the amount of peri-implant bone corticalization after 10 years of functional loading was strongly related to the occurrence of marginal bone loss (*p* < 0.0001) at that time of observation. This is no longer a prediction but only a simulated observation of both treatment outcomes (Figure 4c).

A query of the literature on the structure of the intraosseous portion of the implant did not yield information on a long-term difference in clinical success depending on the features of the implant structure. Even a fairly extensive review of the medical literature in terms of assessing the importance of thread design for success in implant treatment [39] concluded by indicating the need for further research.

It is believed [33] that a valuable question is which features of dental implant design are associated with hyper-corticalization. When evaluating the effect of the titanium alloy used for the implants, it was noted that the grade 4 alloy was associated with less corticalization of the surrounding bone after five and ten years of functional loading. At the same time, grade 4 was associated with less peri-implant bone loss.

Subcrestal- and bone-level-placed implants produced less corticalization than tissue level implants at the ten-year follow-up. In contrast, five-year observations indicated that corticalizing effects and bone loss are lowest with subcrestal implants. This is in full accordance with the observations of the Vilnius Research Group [18,19].

The study confirmed that the current standard abutment with conical or internal hexagon abutments [40] is preferable to one-piece implants or the former internal hexagon because it results in less corticalization in the surrounding environment and little marginal bone loss at the 10-year follow-up.

As can be seen from the comparison of the implant body design, threadless implants were at risk of significant marginal bone loss (exceeding 2 mm over 10 years of use). This was associated with a profound surrounding corticalization over 10 years (BI = 0.41 ± 0.12 and median 0.39). This seems to be a serious warning for designers of future dental implants. As can be seen, a reasonably designed thread (e.g., V-shape or reverse buttress) not only provided primary stability but also protected against unfavorable peri-implant bone remodeling. This seems interesting insofar as it was previously thought that implant thread construction was rather related to achieving only primary stability [41,42]. Similarly, tapered body implants were again considered superior to cylindrical implants due to their easily obtained adequate primary stabilization [43], as one can see the effect of the implant design on the surrounding bone was much more profound and lasts longer than the day of surgery or 2–4 months of observation [32,33].

The presence of microthreads in the implant head, an apex shape as well as a hole in the tip of the implant had no prognostic significance in terms of corticalization and marginal bone loss (excluding the apex hole feature). This confirmed the study of Trisi’s team [42].

By studying the corticalization of the marginal peri-implant bone, a relationship of the structure of the implant apex with this phenomenon was discovered. An apical groove was statistically associated with less bone corticalization. This information was also reinforced by the statistically significant lesser atrophy of the marginal bone. On the other hand, the lack of apical round hole was statistically associated with less bone loss over ten years, although the bone structure did not follow this clinical fact (the BI was the same in both groups of implants, i.e., with round hole and without apical round hole). It is difficult to explain this by the direct influence of the apex on the head of the implant and the corticalization and changes in marginal bone level observed there. It seems that this is an indirect relation or the influence of other features of the structure of implants with apex groove, especially since the greatest structural changes occurred under the influence of forces acting in the cortical substance of the bone [44], i.e., closer to the neck (not the apex) of the dental implant.

A valuable question is which prosthetic restorations cause hypercorticalization. Are there any prosthetic solutions that induce less remodeling in the surrounding bone?

It was tested in a finite element analysis that splinted crowns favor the stress distribution by reducing the stress in the implant/abutment and cortical bone tissue [45]. Relating this to the clinical results obtained, it seems that the numerical experiments did not take into account the role of technical fabrication difficulties (risk of residual stresses, problems with compensating for nonparallelism) and the role of oral hygiene, which is more difficult in splinted crown work. There are reports [46] describing bridgeworks as significantly more affected by marginal crestal bone loss compared to single-crown-supported implants in short-term studies. In this long-term study, however, single crowns were the best prosthetic solution. This seemed to be because a small amount of bone loss occurred in combination with a small degree of corticalization. This seems to be a good prognostic indicator for nonsplinted crowns.

The small marginal bone loss observed with implants loaded with bridgeworks was surprising [47], because the corticalization there was very high (the bone index was in the very low range of 0.39–0.41). Looking at the equally low bone loss in overdentures (unfortunately the group with lowest numbers in this study), it seemed that the large amount of crestal bone between implants neutralized the negative effect of corticalization. There are probably other explanations, because the treatment of people is an extremely multirelated and multithreaded issue that is simultaneously subject to multiple influences. On one hand the periodontal/peri-implant status [48,49] is obviously related to the marginal bone loss (and supposedly to marginal bone remodeling, i.e., corticalization), but on the other hand, the general health condition affects alveolar bone [35,36,50,51,52]. Undoubtedly, the issue of the significance of bone remodeling under dental implant loading requires further study.

However, looking critically at the negative evaluation of the corticalization phenomenon, it seems to pay attention to the physiological mechanisms of bone remodeling as a result of its constant loading (Wolf’s law), where bones subjected to loading presented an increased mineral density [44]. Thus, the conversion of trabecular bone to compact bone is unavoidable. On the one hand it is known [53,54] to lead to complications (bisphosphonian related necrosis of the jaw, BRONJ), such as antiresorptive therapies changing the microarchitecture of the alveolar bone by a medication agent embedding in the mandible, which may subsequently lead to a drug-dependent corticalization and a decrease in the vascularization of the jawbone [55]. On the other hand, it is essential for the success of immediate postextraction implants in patients with normal bone turn-over [33,56,57,58,59,60,61]. Corticalization seems an important but not well-studied phenomenon. It was demonstrated that even after the establishment of osseointegration, bone remodeling could be affected by preload [62], parafunctions, high bending moments [10] or healing-screw-derived topical drag delivery systems [63]. The authors believe that some degree of corticalization is necessary. It remains open to assess how much “some” means. A great deal depends on these future studies that will clarify this nuance, indicate the golden corticalization/trabeculation ratio and other characteristics/features that are not known today.

The study’s limitations are paradoxically its subject matter. The phenomenon of peri-implant bone corticalization requires a further investigation and examination of the numerous detailed treatment factors that may affect bone remodeling locally. Another limitation is the limited number of study participants, despite the evaluation of more than one and a half thousand implants. The last limitation worth mentioning is the follow-up time. Ten years seem like a long time for a planned scientific experiment, but due to the effectiveness of implant treatment, this time is not long for a clinical trial. Many young patients will need implants for decades, not for a single decade.

## 5. Conclusions

The functional loading of intraosseous dental implants caused significant changes in the structure of the alveolar marginal bone, when observed radiographically. There was corticalization and associated marginal bone loss relentlessly progressing over the five and ten years of observation presented here. The conducted analysis strongly suggests that the phenomenon of corticalization is a nonbeneficial alteration of the bone around the implants (at least in the scope disclosed in this study).

## Figures and Tables

**Figure 1 jcm-11-03545-f001:**
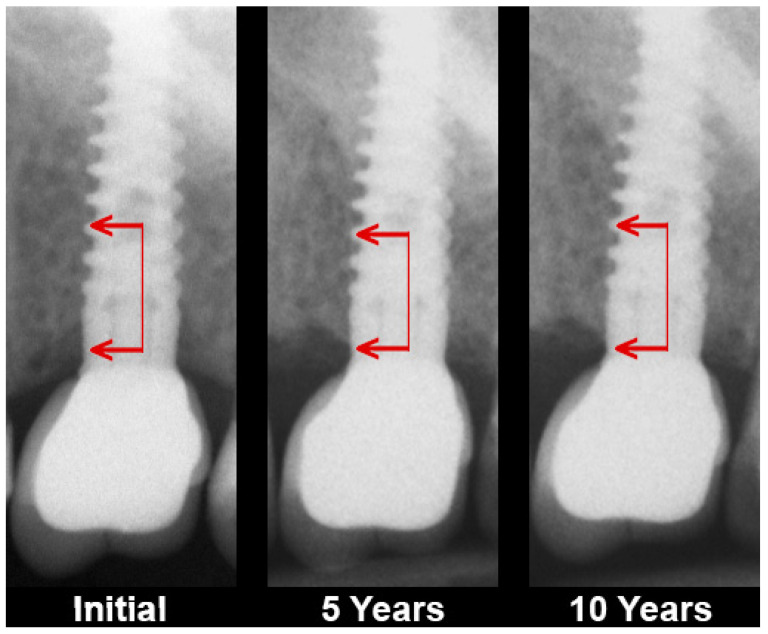
Example of marginal crestal bone alteration 5 and 10 years after the functional loading of a single dental implant by a crown. Bone level fluctuations and the transformation of bone tissue texture at the neck of the dental implant from cancellous to cortical can be seen (i.e., corticalization). The arrows show an example of the extent of peri-implant bone structure transformation.

**Figure 2 jcm-11-03545-f002:**
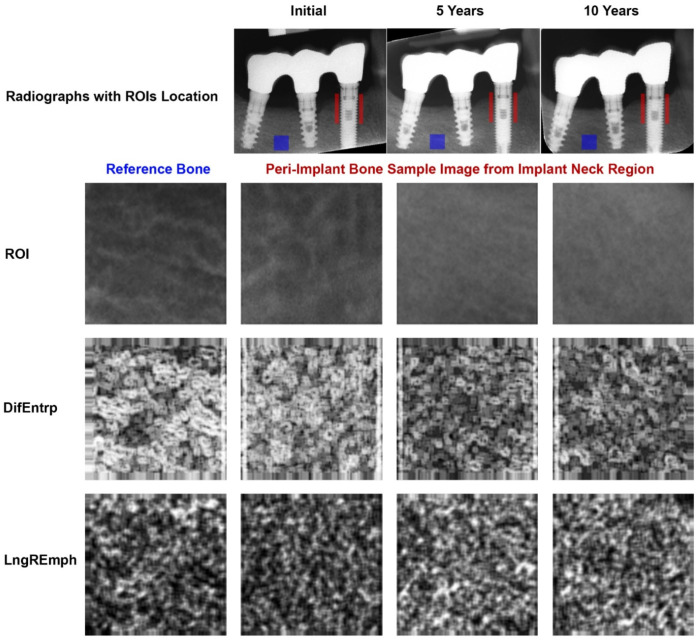
Demonstration of how digital texture analysis was performed on collected radiological material. The column under Initial contains images from the first period of the study, i.e., 3 months after implant insertion, on the day prosthetic work was performed. The next column titled 5 Years contains images from when the implants had been functionally loaded for 5 years. The last column on the right, named 10 Years, contains images from the last examination period, i.e., when the implants had been functionally loaded for 10 years. The top row shows radiographs in a chronological manner. The locations where the data were taken for analysis, i.e., regions of interest (ROIs), are shown here. The ROI of the reference cancellous bone is circled in blue. Two further ROIs (red) are located as close as possible to the dental implants and start at the implant neck and extend 5 mm into the bone. For calculations, data from these two ROIs were combined so that the area of the image sample from the implant neck region was equal to the area of the reference ROI. The second row from the top shows the image samples (ROIs) that were analyzed below (these are still radiological images). The third row from the top shows maps of image texture feature intensity: DifEntrp, from the reference area, through peri-implant at baseline, then after five years of implant loading, and rightmost after 10 years of loading. The whiter areas represent structures with higher entropy (more chaotic structural pattern), and the blacker areas show places where entropy is lower (more regular, homologous structural pattern). The lowest, fourth row of images shows intensity maps of the texture feature: LngREmph. The white areas have more very long structures, while the black areas have few long structures.

**Figure 3 jcm-11-03545-f003:**
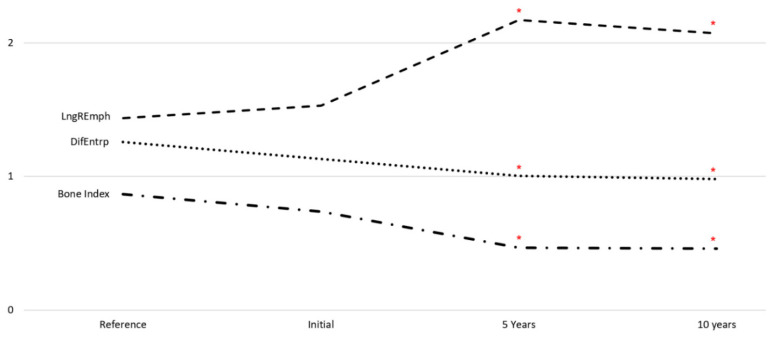
Texture characteristics studied over 10 years (medians). At the time of exposure of the dental implants, all features studied were identical to those of the trabecular reference bone. After five and ten years of functional loading, both studied features of the bone radiograph as well the bone index were statistically significantly different from the reference bone (and from the bone texture at the time of uncovering as well). Red asterisks indicate statistically significant differences from the reference bone; LngREmph—long run-length emphasis moment; DifEntrp—differential entropy.

**Figure 4 jcm-11-03545-f004:**
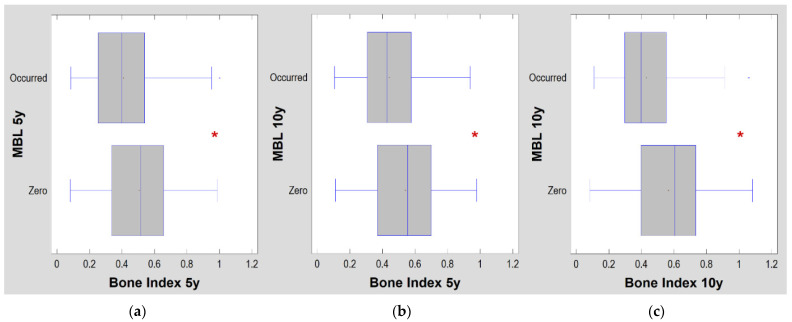
Relationship of crestal bone level to degree of corticalization. Due to the fact that the bone index is a measure of similarity to trabecular (cancellous) bone, therefore, the lower the bone index value, the more corticalized the bone is: (**a**) five-year observations for both variables; (**b**) five-year bone index compared to marginal bone loss noted at the ten-year examination, i.e., predicting whether marginal bone loss will occur after ten years if a given level of bone index value is observed in the fifth year of observation; (**c**) ten-year observations for both variables. The vertical lines in the boxes indicate the median value. Asterisks indicate statistically significant differences. All occurred at the *p* < 0.0001 level. MBL-marginal bone loss; 5y, 10y—five- and ten-year observations.

**Table 1 jcm-11-03545-t001:** Technical features of dental implant designs used in this study. Data were confirmed at www.spotimplant.com/en/dental-implant-identification; accessed on 5 March 2022.

Manufacturer Implant	Titanium Alloy	Level	Connection Type	Connection Shape	Neck Shape	Neck Microthreads	Body Shape	Body Threads	Apex Shape	Apex Hole	Apex Groove
AB Dental Devices I5	Grade 5	Bone level	Internal	Hexagon	Straight	No	Tapered	Square	Flat	No hole	Yes
ADIN Dental Implants Touareg	Grade 5	Bone level	Internal	Hexagon	Straight	Yes	Tapered	Square	Flat	No hole	Yes
Alpha Bio ARRP	Grade 5	Tissue level	Custom	One-piece abutment	Straight	No	Tapered	Reverse buttress	Cone	No hole	No
Alpha Bio ATI	Grade 5	Bone level	Internal	Hexagon	Straight	Yes	Straight	Square	Flat	No hole	Yes
Alpha Bio OCI	Grade 5	Bone level	Internal	Hexagon	Straight	No	Straight	No Threads	Dome	Round	No
Alpha Bio DFI	Grade 5	Bone level	Internal	Hexagon	Straight	Yes	Tapered	Square	Flat	No hole	Yes
Alpha Bio SFB	Grade 5	Bone level	Internal	Hexagon	Straight	No	Tapered	V-shaped	Flat	No hole	Yes
Alpha Bio SPI	Grade 5	Bone level	Internal	Hexagon	Straight	Yes	Tapered	Square	Flat	No hole	Yes
Argon Medical Prod. K3pro Rapid	Grade 4	Subcrestal	Internal	Conical	Straight	Yes	Tapered	V-shaped	Dome	No hole	Yes
Bego Semados RI	Grade 4	Bone level	Internal	Hexagon	Straight	Yes	Tapered	Reverse buttress	Cone	No hole	Yes
Dentium Super Line	Grade 5	Bone level	Internal	Conical	Straight	No	Tapered	Buttress	Dome	No hole	Yes
Friadent Ankylos C/X	Grade 4	Subcrestal	Internal	Conical	Straight	No	Tapered	V-shaped	Dome	No hole	Yes
Implant Direct InterActive	Grade 5	Bone level	Internal	Conical	Straight	Yes	Tapered	Reverse buttress	Dome	No hole	Yes
Implant Direct Legacy 3	Grade 5	Bone level	Internal	Hexagon	Straight	Yes	Tapered	Reverse buttress	Dome	No hole	Yes
MIS BioCom M4	Grade 5	Bone level	Internal	Hexagon	Straight	No	Straight	V-shaped	Flat	No hole	Yes
MIS C1	Grade 5	Bone level	Internal	Conical	Straight	Yes	Tapered	Reverse buttress	Dome	No hole	Yes
MIS Seven	Grade 5	Bone level	Internal	Hexagon	Straight	Yes	Tapered	Reverse buttress	Dome	No hole	Yes
MIS UNO One Piece	Grade 5	Tissue level	Custom	One-piece abutment	Straight	No	Tapered	Square	Dome	No hole	Yes
Osstem Implant Company GS III	Grade 5	Bone level	Internal	Conical	Straight	Yes	Tapered	V-shaped	Dome	No hole	Yes
SGS Dental P7N	Grade 5	Bone level	Internal	Hexagon	Straight	Yes	Tapered	V-shaped	Flat	No hole	Yes
TBR Implanté	Grade 5	Bone level	Internal	Octagon	Straight	No	Straight	No threads	Flat	Round	Yes
Wolf Dental Conical Screw-Type	Grade 4	Bone level	Internal	Hexagon	Straight	No	Tapered	V-shaped	Cone	No Hole	Yes

**Table 2 jcm-11-03545-t002:** Pooled data describing the progressive increase in the difference in bone structure of implant-loaded versus reference trabecular bone (bone index) and the observed associated, marginal bone loss (MBL).

Region of Interest/Period	Bone Index	Marginal Bone Loss (mm)	Simple Regression
Reference cancellous site	0.85 ± 0.18	n.a.	n.a.
Initial peri-Implant observation	0.73 ± 0.21	0.00 ± 0.00	n.a.
5-year peri-implant observation	0.47 ± 0.21	0.88 ± 1.27	CC = −0.26; R^2^ = 7%; *p* < 0.0001
10-year peri-implant observation	0.48 ± 0.21	1.49 ± 1.94	CC = −0.28; R^2^ = 8%; *p* < 0.0001

Abbreviations: n.a.—not applicable; CC—correlation coefficient.

**Table 3 jcm-11-03545-t003:** Peri-implant bone feature observed among examined implant designs in this study.

Design Parameter	Option	Feature	Initial	5 Years	10 Years
Titanium alloy *n* = 1447	Grade 4	MBL	0.00	0.00 ^L^	0.00 ^L^
		BI	0.74	0.66 ^H^	0.64 ^H^
	Grade 5	MBL	0.00	0.00 ^H^	1.09 ^H^
		BI	0.74	0.46 ^L^	0.46 ^L^
Immersion level *n* = 1275	Subcrestal	MBL	0.00	0.00 ^L^	0.00 ^L^
		BI	0.68	0.68 ^H^	0.69 ^H^
	Bone level	MBL	0.00	0.00 ^H^	1.09 ^H^
		BI	0.74	0.46 ^L^	0.46 ^H^
	Tissue level	MBL	0.00	1.33 ^H^	0.48 ^H^
		BI	0.68	0.36 ^L^	0.20 ^L^
Connection type *n* = 1275	Internal	MBL	0.00	0.00	1.09
		BI	0.74	0.47	0.46 ^H^
	Custom	MBL	0.00	1.33	0.48
		BI	0.68	0.36	0.20 ^L^
Connection shape *n* = 1275	Conical	MBL	0.00	0.00	0.00 ^L^
		BI	0.67	0.50	0.54 ^H^
	Internal hexagon	MBL	0.00	0.00	1.09
		BI	0.76 ^H^	0.46	0.46 ^H^
	Internal octagon	MBL	0.00	0.67	2.91 ^H^
		BI	0.45 ^L^	0.44	0.32
	One-piece abutm.	MBL	0.00	1.33	0.00 ^L^
		BI	0.68	0.36	0.20 ^L^
Head microthreads *n* = 1275	Yes	MBL	0.00	0.00	0.87
		BI	0.73	0.47	0.47
	No	MBL	0.00	0.61	1.15
		BI	0.77	0.45	0.42
Body shape *n* = 1447	Tapered	MBL	0.00	0.00 ^L^	0.91 ^L^
		BI	0.73 ^L^	0.46 ^L^	0.47
	Straight	MBL	0.00	1.57 ^H^	1.82 ^H^
		BI	0.82 ^H^	0.56 ^H^	0.44
Body threads *n* = 1447	Buttress	MBL	0.00	2.15 ^H^	n.a.
		BI	0.32 ^L^	0.21 ^L^	n.a.
	Reverse buttress	MBL	0.00	0.00 ^L^	0.79
		BI	0.72	0.45	0.47
	V-shape	MBL	0.00	0.00 ^L^	0.00 ^L^
		BI	0.78 ^H^	0.58 ^H^	0.56 ^H^
	Square	MBL	0.00	0.39	1.21 ^H^
		BI	0.76	0.47	0.45
	No threads	MBL	0.00	1.24	2.42 ^H^
		BI	0.69	0.51	0.39 ^L^
Apex shape *n* = 1447	Cone	MBL	0.00	0.00	0.00
		BI	0.92 ^H^	0.53	0.53
	Dome	MBL	0.00	0.00 ^L^	0.79
		BI	0.72 ^L^	0.46	0.46
	Flat	MBL	0.00	0.45 ^H^	1.21
		BI	0.79 ^H^	0.47	0.45
Apex hole *n* = 1447	Round	MBL	0.00	1.24	2.42 ^H^
		BI	0.69	0.47	0.39
	No or other	MBL	0.00	0.00	0.97 ^L^
		BI	0.73	0.51	0.47
Apex groove *n* = 1275	Yes	MBL	0.00	0.00 ^L^	0.98 ^L^
		BI	0.74	0.47	0.46 ^H^
	No	MBL	0.00	1.63 ^H^	1.69 ^H^
		BI	0.66	0.31	0.37 ^L^

^H^ value higher than in other implant design options within observation period (*p* < 0.05); ^L^ value lower than in other implant design options within observation period (*p* < 0.05); *n*—number of evaluated dental implants; MBL—marginal bone loss is given as median due to non-normal distribution in mm; BI—bone index is given as median due to non-normal distribution.

**Table 4 jcm-11-03545-t004:** The prosthetic solutions examined in this study.

Prosthetic	*n*	Feature	Initial	5 Years	10 Years
Single crown	493	MBL	0.00	0.00	0.90
		BI	0.78 ^H^	0.53 ^H^	0.55 ^H^
Splinted crowns	510	MBL	0.00	0.08	1.29 ^H^
		BI	0.73	0.46	0.44 ^L^
Bridge	384	MBL	0.00	0.00 ^L^	0.00 ^L^
		BI	0.69 ^L^	0.39 ^L^	0.41 ^L^
Overdenture	89	MBL	0.00	0.48	0.00 ^L^
		BI	0.65 ^L^	0.30 ^L^	0.31
Platform switching	383	MBL	0.00	0.00	0.00
		BI	0.67 ^L^	0.48	0.56

^H^ value higher than in other prosthetic solutions (*p* < 0.05); ^L^ value lower than in other prosthetic solutions (*p* < 0.05); *n*—number of evaluated dental implants; MBL—marginal bone loss is given as median due to non-normal distribution in mm; BI—bone index is given as median due to non-normal distribution.

## Data Availability

The data on which this study is based will be made available upon request at https://www.researchgate.net/profile/Marcin-Kozakiewicz, access date: 10 June 2022.

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
