# Peer review of "What Does Bone Corticalization around Dental Implants Mean in Light of Ten Years of Follow-Up?"

_jcm, 2022, doi:10.3390/jcm11123545_

Round 1
Reviewer 1 Report
This study is a prospective study that attempts to analyze the clinical significance of peri-implant bone corticalization after functional loading. This study included 554 patients, and 1556 implants, with a significant follow-up period of 10 years of functional loading.
Although an intriguing subject the manuscript has several issues that must be addressed.
Please see the enclosed PDF for further details.

Author Response
Please, find the attached file with detailed reply.

Reviewer 2 Report
Dear Authors,
thank you for your manuscript dealing with bone corticalization around dental implants. I think that your work is really interesting and with good scientific soundness. The radiologic documentation is very relevant.
Howver, the only issue in my opinion is related to the bibliography. I think that you should remove old references and add new ones. Moreover, it could be interesting to add in the discussion the influence of the treatment of peri-implant sites on the parameters evaluated by you. In particular, the administration of ozone represent a hot topic. You could for example refer to the following research.
Butera, A.; Gallo, S.; Pascadopoli, M.; Luraghi, G.; Scribante, A. Ozonized Water Administration in Peri-Implant Mucositis Sites: A Randomized Clinical Trial. Appl. Sci. 2021, 11, 7812. https://doi.org/10.3390/app11177812
Yours faithfully,
The Reviewer
Author Response

(The authors gave the same response as above.)

Round 2
Reviewer 1 Report
The authors have improved the manuscript.
Author Response
Thank you for your appreciation.
Best regards
Marcin Kozakiewicz
Reviewer 2 Report
Dear Authors,
congratulations for your work. You have addressed all the points suggested by me.
There is a last thing which I think could be still done to improve the quality of your manuscript which is, if possible, an improvement of the quality of figure 2.
Anyway the manuscript now deserves to be published in the journal.
Yours faithfully,
The Reviewer
Author Response
Dear Sir
Thank you for time and advises. I finally have improved fig. 2 and fig3 in the latest version of revision.